# Relationship between CT-Derived Bone Mineral Density and UTE-MR-Derived Porosity Index in Equine Third Metacarpal and Metatarsal Bones

**DOI:** 10.3390/ani13172780

**Published:** 2023-08-31

**Authors:** Carola Riccarda Daniel, Sarah Elizabeth Taylor, Samuel McPhee, Uwe Wolfram, Tobias Schwarz, Stefan Sommer, Lucy E. Kershaw

**Affiliations:** 1Royal (Dick) School of Veterinary Studies, The Roslin Institute, The University of Edinburgh, Edinburgh EH25 9RG, UK; 2Institute of Mechanical, Process and Energy Engineering, School of Engineering and Physical Sciences, Heriot-Watt University, Edinburgh EH14 4AS, UK; sm113@hw.ac.uk (S.M.);; 3Siemens Healthcare, 8047 Zurich, Switzerland; sommer.stefan@siemens-healthineers.com; 4Swiss Center for Musculoskeletal Imaging (SCMI), Balgrist Campus, 8008 Zurich, Switzerland; 5Advanced Clinical Imaging Technology (ACIT), Siemens Healthcare AG, 1015 Lausanne, Switzerland; 6Centre for Cardiovascular Sciences and Edinburgh Imaging, The University of Edinburgh, Edinburgh EH16 4TJ, UK; lucy.kershaw@ed.ac.uk

**Keywords:** BMD, PI, MRI, ultra-short echo time, CT, equine, lateral condylar fracture, palmar osteochondral disease, stress fracture, fatigue injury

## Abstract

**Simple Summary:**

Racehorses have similar health issues to human runners during their careers. The intense exercise regime leads to changes in their bone compositions where their bones become more dense and less porous in order to adapt to the higher-than-usual physical demands. While this is generally beneficial, their bones also become more brittle in the process, and this can result in stress fractures in human athletes and racehorses alike. The balance between beneficial training responses and overtraining is fine, and we are currently not able to use imaging techniques to distinguish reliably between the two of them before injuries occur. Two important markers of bone health are the density of the bone, which is well established, and the number of pores within the bone tissue, and the latter has recently attracted more interest. We aimed to validate a new MRI-based technique for measuring bone porosity in horses. This has the potential to improve our understanding of bony changes associated with training in racehorses and aid in the identification of factors associated with fracture risks.

**Abstract:**

Fatigue-related subchondral bone injuries of the third metacarpal/metatarsal (McIII/MtIII) bones are common causes of wastage, and they are welfare concerns in racehorses. A better understanding of bone health and strength would improve animal welfare and be of benefit for the racing industry. The porosity index (PI) is an indirect measure of osseous pore size and number in bones, and it is therefore an interesting indicator of bone strength. MRI of compact bone using traditional methods, even with short echo times, fail to generate enough signal to assess bone architecture as water protons are tightly bound. Ultra-short echo time (UTE) sequences aim to increase the amount of signal detected in equine McIII/MtIII condyles. Cadaver specimens were imaged using a novel dual-echo UTE MRI technique, and PI was calculated and validated against quantitative CT-derived bone mineral density (BMD) measures. BMD and PI are inversely correlated in equine distal Mc/MtIII bone, with a weak mean r value of −0.29. There is a statistically significant difference in r values between the forelimbs and hindlimbs. Further work is needed to assess how correlation patterns behave in different areas of bone and to evaluate PI in horses with and without clinically relevant stress injuries.

## 1. Introduction

Racehorses experience high volumes of intense exercise, resulting in the increased cyclic loading of their musculoskeletal systems and increased risks of stress injuries compared to the general horse population. The most commonly reported fatigue injuries in racehorses are palmar/plantar osteochondral disease (POD) [1] and fractures of the third metacarpal and metatarsal bones (McIII and MtIII) [2].

POD is a common condition affecting the subchondral bone of the palmar and plantar distal condyles of McIII and MtIII. The condition is usually most pronounced in the thoracic limbs at the medial condyles; however, the pelvic limbs can also be affected, where the lateral condyles are predominantly diseased [3]. In addition to subchondral bone lesions, cartilage ulceration and wear lines can be found in affected horses [1]. Lesion severity ranges from discoloration of the subchondral bone to disruption of the osseous and cartilaginous surfaces, creating full thickness focal defects [1,4]. Pain associated with subchondral bone injury is well-documented and associated with performance limiting lameness in the equine patient [5,6]. Radiographic assessments of POD lesions have shown low sensitivity and specificity but can be aided by secondary features of advanced disease such as osteophytes on the proximal sesamoid bones, cavitation of the dorso-distal aspect of the McIII bone, or flattening of the palmar condyles [7]. An in vivo standing low-field MRI is able to identify osseous changes associated with palmar osteochondral disease prior to radiographic changes [8], and a high-field MRI can assess cartilaginous involvement [9], but the in vivo differentiation of physiological rather than pathological subchondral bone changes, as seen in the early stages of the disease, remains difficult. To avoid progression of disease to debilitating stages, early detection would be beneficial. Furthermore, in some cases, stress-induced subchondral bone injuries have the potential to propagate into potentially catastrophic fractures [10,11].

Fractures of the McIII and MtIII bone are common injuries in racehorses and are associated with animal welfare concerns [12,13] and financial losses [14]. The most common fracture site is the lateral condyle of the McIII or MtIII bone emanating from or near the parasagittal groove [2,13]. These fractures are classified as fatigue fractures [15] and represent a known clinical problem in human and equine athletes alike [16,17]. In these patient groups, repetitive and excessive strain from high-volume training leads to the accumulation of microtrauma, which, over time, reduces bone stiffness and decreases bone strength [18,19]. Eventually, the accumulation of microdamage is greater than the reparative capacity of the osseous tissues, and catastrophic fractures can occur. While the direct assessment of accumulative osseous microdamage represents an interesting tool for investigating fracture risks, it is currently limited to experimental ex vivo imaging as suitable and direct in vivo imaging modalities are missing [17,20]. Furthermore, commercially available microCT scanners only allow for the imaging of small samples due to size restrictions. The assessment of a whole bone as an uninterrupted anatomical and biomechanical unit is unattainable with a microCT scanner. As an indirect measure of microdamage and repetitive stress injuries, the assessment of bone marrow oedema-type signals on low-field MRI has been described in equine patients [21]. The use of nuclear medicine (scintigraphy) has also been described as useful tool for the in vivo early detection of stress injuries in humans, with sensitivities ranging from 74 to 100% in the literature [22,23,24].

Bone mineral density (BMD) is widely used in human medicine as an indirect measure of fracture risk, particularly in the context of fragility fractures and osteoporosis [25]. While traditionally measured via dual-energy X-ray absorptiometry (DEXA), the use of quantitative computed tomography (QCT) has recently come to the forefront [26]. For equine patients, QCT has shown greater BMD values in response to race training in the diaphyseal and condylar portions of McIII bone; however, no significant differences between fracture patients and controls were identified in two past studies [27,28]. However, the results of work by Loughridge et al. (2017) were at variance with previous studies, showing significantly higher bone densities in the fractured McIII bones of racehorses when compared to the controls [29]. The use of QCT-derived BMD measurements remains of interest in the clinical context as it allows the use of whole, undissected specimens as well as in vivo data derived from clinical patients.

In addition to the mineral component, the porosity of cortical and trabecular bone plays a major role in bone strength and stiffness [30,31], and has been associated with fractures—independent of bone density—in humans [32,33]. While, traditionally, bone and bone porosity have been assessed with radiation-based modalities, MRI has recently enjoyed increasing popularity. Ultra-short echo time (UTE) MRI is a novel technique capable of imaging cortical bone [34,35,36]. In conventional MR sequences, the signal decay times of cortical bone are too short to generate detectable signal, but the echo times in UTE sequences allow sufficient signal detection to image cortical bone even with its tightly bound water protons. Previous efforts have focused on simply reducing the echo time (TE) in bone imaging, but recently, the focus has shifted towards differentiating between the following two pools of labile water protons: bound water (BW) and pore water (PW) [37]. Porous water (PW), which can move freely within osseous pores, has a longer T2* relaxation time than collagen-bound water (BW). MRI signal detected from PW is an indirect measure of the amount of pore space within cortical bone, and has been shown to be indirectly proportional to bending strength and to agree with microCT-derived porosity data [38,39]. The porosity index (PI) is defined as the ratio between PW and BW and can be computed by using a dual-echo technique, acquiring PW as well as BW images. For this technique, two images are acquired: one with a relatively long echo time, which solely contains signals from PW as BW signals have fully decayed at read out, and the other uses a relatively short echo time and will therefore include signals from both water proton pools. Dividing the long echo time image by the short echo time image will result in a PI image [38,40]. Strong correlations between microCT-derived porosity indexes, pore sizes, and bone stiffness levels have been described in ex vivo experiments and in human patients [38,40,41].

We aimed to validate the use of a dual-echo UTE research application sequence for the measurement of PI in the equine patient against a peripheral quantitative CT (QCT). Furthermore, we hypothesized that BMD and PI are inversely correlated in the distal McIII/MtIIl bone of racehorses in training, in keeping with known data from human patients. We further aimed to investigate the potential differences between forelimbs and hindlimbs and the left and right sides. PI may serve as an indicator of bone health and strength and aid our understanding of the pathophysiological processes in the context of osseous stress remodeling and fatigue injuries in equine athletes.

## 2. Materials and Methods

### 2.1. Animals

Specimens were derived from six thoroughbred racehorses in training (aged 3 to 9 years, 2 mares and 4 geldings). All animals were subjected to euthanasia between November 2021 and April 2022 for non-study related reasons, and their limbs were donated to research (VERC 165.22). Twelve forelimbs and twelve hindlimbs were harvested within 24 h of the deaths of the animals. The specimens were disarticulated in the carpal and tarsal joints, respectively, and frozen at −20° for up to nine months (ranging from 40 days to 282 days) until further processing. All specimens were brought to room temperature over a period of 24 h prior to the CT and MR imaging.

### 2.2. CT Imaging

All specimens underwent high-resolution peripheral quantitative computed tomography in a 64-slice fan beam CT scanner (SOMATOMDefinition AS, Siemens Healthcare, Erlangen, Germany; voltage: 120 kV, current: 80 mA, matrix size: 512 × 512, slice thickness: 0.6 mm, and voxel size of 0.273 mm × 0.273 mm × 0.4 mm). The limbs were scanned with a standard clinical hydroxyapatite phantom (model number 8783219, Siemens, Erlangen, Germany) to allow for calibration to a bone mineral density scale.

### 2.3. MR Imaging

Three-dimensional dual-echo UTE images were acquired using a 3T clinical scanner (MAGNETOM Skyra 3T, Siemens Healthcare, Erlangen, Germany; short echo time (TE_short_): 0.04 ms, long echo time (TE_long_): 2.52 ms, repetition time (TR): 12 ms, fat suppression, average voxel size: 0.66 mm^3^ isotropic, matrix size 304 × 304, and field of view (FOV) 200 mm × 200 mm). Specimens were positioned in a 15-channel transmit/receive (Tx/Rx) knee coil with the dorsal aspect of the fetlock pointing up and the distal aspect of the limb pointing out of the machine (human equivalent positioning: head-first supine (HFS). All specimens were wrapped in a plastic specimen bag for hygiene reasons. The UTE images were acquired in a frontal plane with isotropic voxels.

### 2.4. Image Processing

The raw 16-bit grey-level QCT images were parsed to a BMD scale by linear transformation based on the hydroxyapatite/demineralized water phantoms using ImageJ (NIH Image J 1.53) [42,43]. PI images were generated by dividing the echo intensity images acquired with the TE_long_ by the echo intensity of images acquired with the TE_short_ and multiplying the result by 100% (see Figure 1) using python.

To spatially align the CT and PI images, the PI image was registered to the CT image using Elastix software (Version v5.0.1 [44,45]). First, a manual rigid alignment was performed, followed by a multi-resolution affine registration using a mutual information metric. The resultant registered PI image matched the resolution of the CT image, allowing for voxel-to-voxel comparison.

The volumes of the distal third metacarpal/metatarsal bone were initially segmented from the QCT images following the process used by McPhee et al. (2023) [42] and utilizing ImageJ software (NIH Image J 1.53 [43]). First, noise was removed by a median filter (kernel size of 2) before being converted to binary using a contrast-based local thresholding filter (kernel size of 50). Finally, the third metacarpal/metatarsal bone was isolated from the background voxels using a connectivity filter.

Co-registered images were loaded into an open-source 3D imaging software (Slicer Version 5.4.0 [46]), and the diaphyseal part of the segmentation of the third metacarpal/metatarsal bone was manually removed to match the ROI to the distal condyle of the third metacarpal/metatarsal bone (see Figure 2).

### 2.5. Statistics

Linear regression analysis was performed to determine the relationship between the BMD and PI. Data were tested for normality using a Jarque–Bera test, and subsequently, a *t*-test was used to assess the differences in r value between left and right limbs, forelimbs, and hindlimbs, as well as left and right forelimbs and left and right hindlimbs. A *p*-value of less than 0.05 was considered statistically significant.

## 3. Results

### 3.1. Imaging

All collected specimens underwent CT imaging. One forelimb was excluded from the study due to the presence of metallic surgical implants, which precluded its use for MR imaging and would have negatively impacted CT images as well as image processing. Therefore, eleven forelimbs and twelve hindlimbs were included in the final analysis.

### 3.2. Linear Regression Anlysis

For each limb, the slope and r value were highly statistically significant different from zero, with *p* values of less than 0.0000. This confirmed a significant linear relationship between BMD and PI, and linear regression analysis was suitable for modeling the linear relationship between the two parameters.

Linear regression analysis revealed a negative correlation between the BMD and PI, with a weak mean r value of −0.29 (ranging from −0.14 to −0.43; SD: 0.071 and SE: 0.014) [47] and a mean slope of −0.0084 (ranging from −0.012 to −0.0048; SD: 0.0018 and SE: 0.00038; for an example, see Figure 3).

### 3.3. Inter-Limb Comparison

A stronger negative correlation between BMD and PI values was observed in the hindlimbs (mean r value of −0.32) when compared to forelimbs (mean r value of −0.26), which reached statistical significance (*p* = 0.036). The remaining comparisons between left (mean r value of −0.28) and right limbs (mean r value of −0.30), left forelimbs (mean r value of −0.25) and right forelimbs (mean r value of −0.26), and left hindlimbs (mean r value of −0.30) and right hindlimbs (mean r value of −0.34) were not statistically significant, with *p* values of 0.39, 0.94, and 0.23, respectively. An overview of all obtained r values can be seen in Table 1.

## 4. Discussion

The technical feasibility of MR-derived porosity was demonstrated in equine cadaver limbs. This is in line with work in human tibiae, which evaluated the technique against microCT measurements of pore size and porosity [40]. The use of UTE MRI and PI for bone imaging has several practical advantages. Firstly, studies can be carried out using a clinical MRI scanner with a standard knee coil. This allows for scanning intact, undissected specimens and assessing the anatomical structures as a whole, rather than as isolated parts, with small samples as needed for other modalities such as microCT. Additionally, despite high-field MRI scanners currently not being standard in large animal hospitals, the availability of standard clinical equipment may still supersede the availability of more specialized research equipment such as microCT or MRI of ultrahigh field strengths. In addition to the use of commercially available standard hardware, the software used in this study is also widely accessible. While a work-in-progress (WIP) dual-echo sequence was used for image acquisition in the current work, similar sequences are readily available on the market. The mathematics behind generating the PI images relied on a simple ratio between two images of different echo times, and they can therefore be computed with free open-source image processing software and do not require specialized calibrations against phantoms or the use of adiabatic inversion pulses as other markers to be utilized in the quantitative MR bone imaging [37]. UTE and zero-echo-time (ZTE) MRI has been used in low-field systems [48,49], and while this is currently still in an experimental state, it demonstrates their technical feasibility. If developed further, this could make the technique highly relevant to equine MR imaging in the future as low-field standing systems are widely used.

Besides the practical advantages of using standard clinical equipment, there are also scientific benefits. The use of a 3T magnet is advantageous over ultra-high field strength research scanners as the susceptibility artefact between water and bone is more pronounced with higher field strengths, which impairs the separation of the BW and PW signals, making PI assessments more difficult [50]. It has been shown that UTE MRI-derived bone porosity measurements are excellent indirect measures of bone microstructure in humans as they are capable of generating signal from all water protons within bones, including those located in micropores even beyond the resolution of microCT images [51]. Osseous microstructure in mammalian species is similar [52,53], and the technical feasibility of UTE MRI for equine bone has been demonstrated in this study, making UTE MRI-derived PI an interesting field for future research.

For the present work, cadaver specimens have been used, which had been frozen and were thawed at room temperature prior to imaging. The use of freezing for specimen preservation has been shown to be superior to formalin-based fixation techniques for MR imaging and did not affect image quality [54]. Specimens were thawed at room temperature, which is preferable over the use of heat mats [54] and prolonged thawing at 4 °C [55]. It has been reported in the literature that the signal intensity of bone marrow can be altered by the subjective hyperintensity in turbo spin-echo T2-weighted and proton density images after thawing at room temperature [55]. However, other authors have found no clinical or statistical differences in the short-tau inversion-recovery (STIR) signal intensity in thawed equine fetlocks [56]. While no studies have explored the effects of freezing and thawing on UTE sequences specifically, the use of frozen and thawed human cadaver specimens—both whole bones and sections—is common in this research field and widely accepted [34,38,41,51].

The entire volume of the metacarpal/metatarsal condyle was chosen as the region of interest in this study. This decision was governed by anatomical, as well as technical, considerations. Recent human studies have mainly focused on the PI of the cortical bone [38,40,57]. With cortical bone being the main contributor to bone strength [38,58,59], this focus appears justified. However, it has been shown that trabecular bone also contribute to strength and stiffness [31], which is relevant to the equine condyle, which contains a large amount of trabecular bone. Additionally, the anatomical porosity of cortical and trabecular bone has to be understood as a closely linked continuum rather than as completely separate components [58]. Therefore, assessments of the entire condyle as an anatomical and biomechanical unit seems to be indicated. However, it cannot be ignored that the calculated numeral porosity value of the two bone compartments was not related. Cortical bone in human mandibular condyles showed an average porosity of 3.5% as compared to trabecular bone, which has an average porosity of 79.3%; additionally, adaptive remodeling has been shown to be different between cortical and trabecular bone [60]. Therefore, careful segmentation of cortical and trabecular bone could aid better understanding the details of the pathophysiological processes of fatigue injuries in equine athletes. Due to the thin cortical bone coverage in the metacarpal/metatarsal equine condyles and the associated partial volume artefacts, segmentation has proven challenging. In clinical patients, this problem could be exacerbated as the equine condyle densify in response to training and are replaced by compact bone, which is akin to cortical bone in density [29,61]. Separation between cortical bone and densified and physiological trabecular bone can, therefore, not easily be achieved. Further work is needed to investigate the full potential of using the entire condyle as compared to condylar segments.

The r values calculated in this study predominantly suggested a weak negative correlation between the BMD and PI, and two specimens reached moderate correlations, with r values of −0.43 and −0.40. This was in contrast to previous work that used cadaveric human femora, where a strong negative correlation with an r value of −0.64 [38] was found. The reasons for this are not fully understood; however, the work by Jones et al. (2021) [38] focused on cortical bone only, while in the present study, a mixed population of trabecular and cortical bone was used, which could be responsible for the differences in the correlation strength.

It should also be noted that assessing bone porosity with a UTE MRI has been described in humans in the context of osteoporosis research. In one study, the osteoporosis status of the used cadaver specimens was not been recorded, but the average age of the included individuals was 72.13 years (range of 44 to 93 years) [38]. This population allows limited comparisons in terms of bone composition when compared to younger individuals [62,63]. The impact of the differences in the ages of the target populations is unknown; however, these differences may have contributed to the contrasting correlation strengths between the PI and BMD values.

It was also noted that a large variability between the r values was present, with values ranging from −0.16 to −0.43. A partial explanation for this could be offered by the varying prevalence and severity of fatigue injury between forelimbs and hindlimbs in the literature. Pinchbeck et al. (2013) stated that POD is more pronounced in forelimbs when compared to hindlimbs in UK-based racehorses [3]. Similarly, 72% of lateral condylar fractures have occurred in forelimbs in the UK [2]. While the clinical relevance and correlation to bone strength of this finding is currently unknown, it is in keeping with the statistically significant difference in the r values between the forelimbs and hindlimbs found in this study. Another possible contributor to the large variability in r values could be laterality as the inside limb was affected in 47% of British clinical cases [2]. No statistically significant difference was found between the left and right limbs in the present work. However, the sample size was small and not only race direction but also training direction needs to be taken into account, and no training records were accessible for the horses in this study.

This study had several limitations, including the small sample size, lack of training records for the included horses, and the use of cadaver specimens. Additionally, no correlations with biomechanical testing could be provided. Mechanical testing could shed light on the relationship between the PI of the combined trabecular and cortical components of the equine distal condyle and bone strength. Alternatively, the use of a limb-specific finite element model, as developed by S.MP. and U.W., could be used for further investigation into condylar PI and bone strength [42]. Segmentation of the region of interest could complement the dataset. Segmentation of cortical and trabecular bone, as well as a focus on the specific locations of fatigue injuries, could be of interest. The behavior of PI in these disease-prone regions would improve the understanding of PI as a biomarker of bone strength. This could be of interest as a novel way to assess fracture risk in the equine athlete as an association between the CT-derived BMD and injury has not been reliably established for condylar fractures in racehorses [27,28,29]. In human bones, cortical PI has been shown to be a chief contributor to bone strength, which is related to fracture risk independent of BMD [64]. However, these studies have been derived from human osteoporosis patients, and their translational value is limited. Data on PI in the context of increased fatigue fracture risk in humans or horses have not been reported in the literature.

## 5. Conclusions

UTE-MRI-derived PI measurements are promising biomarkers for assessing bone health and strength. A better understanding of osseous porosity and its correlations with other biomarkers, as well as disease patterns, could improve our understanding of the pathophysiological processes associated with fatigue injuries and their presentation in clinical imaging modalities.

This proof-of-concept study was able to establish the feasibility of the use of UTE-MRI in equine specimens, and it established an inverse correlation between BMD and condylar PI values in equine distal McIII/MtIII bone. Further work is needed to segment the equine condyle into its different osseous components and to evaluate PI in horses with and without clinically relevant stress injuries in the metacarpal/-tarsal condyle, as well as to correlate these patterns to training history gathered with GPS data.

## Figures and Tables

**Figure 1 animals-13-02780-f001:**
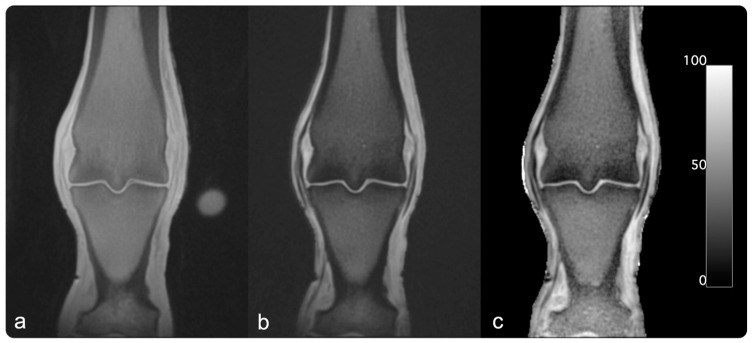
Example MR images: UTE images were acquired using dual-echo UTE sequences. Image (**a**) was acquired with a short echo time of 0.04 ms (*echo*_1_). Image (**b**) was acquired with a longer echo time of 2.68 ms (*echo*_2_). For the PI image in panel (**c**), porosity index was calculated using the following formula: $Porosity\;index\;\left(\%\right)=\frac{{intensity\;echo}_{2}}{{intensity\;echo}_{1}}\times 100\%$. For presentation in this figure, the background was manually set to zero.

**Figure 2 animals-13-02780-f002:**
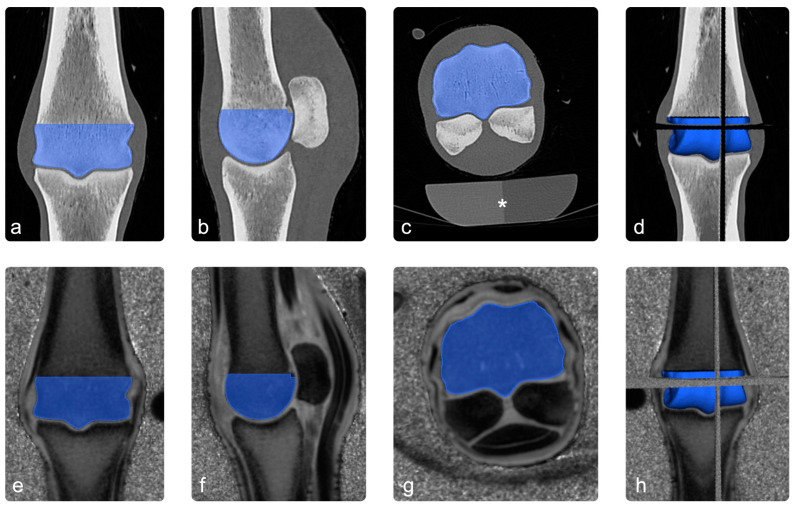
Image processing: images (**a**–**c**) show the ROI (blue) in the coronal, sagittal, and transverse sections of the CT scan. In image (**d**), the three planes were combined to a 3D-ROI. Images (**e**–**h**) show the co-registered PI images in the same way. The asterisk in image (**c**) highlights a standard osteodensitometry phantom containing calcium hydroxyapatite and demineralized water, which was used to calculate bone-mineral density.

**Figure 3 animals-13-02780-f003:**
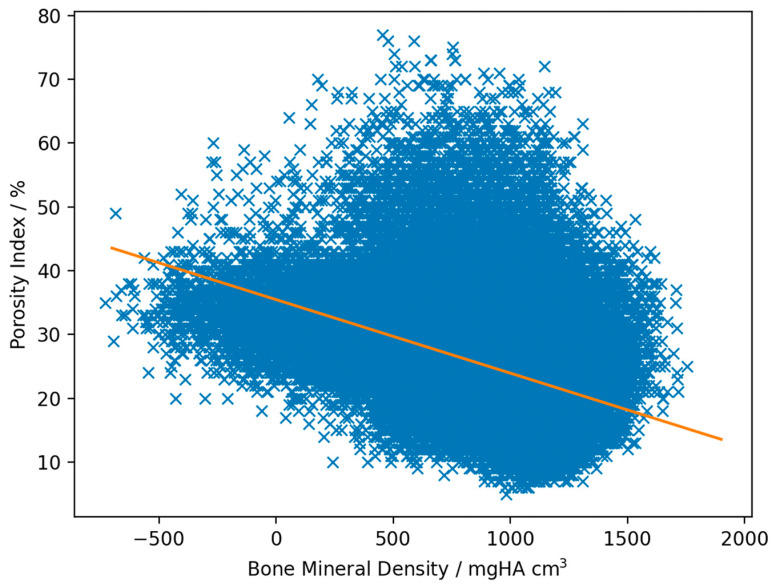
Scatterplot example of one specimen. The volume of the distal third metacarpal/metatarsal bone was initially segmented from the QCT image following the process used by McPhee et al. (2023) [42]. The whole metacarpal/metatarsal condyle was included in the 3D-ROI. Care was taken to not include the soft tissues surrounding the osseous structures. The same ROI was used in the co-registered MR images. The BMD values derived from CT images and PI image derived from MR images were plotted against each other for each pixel. The negative slope of −0.0087 indicated an inverse correlation between the two variables. The r value of the plotted specimen was −0.43.

**Table 1 animals-13-02780-t001:** Slope, standard error of the slope, and r values are listed per horse and per limb, as used in the study. For all specimens, the slopes and correlation coefficients (r values) were statistically significant different from zero, with *p* values of less than 0.00001. The right forelimb of horse 1 was excluded due to the presence of metallic surgical implants. *, very weak correlation; **, weak correlation; ***, moderate correlation; LF, left forelimb; RF, right forelimb; LH, left hindlimb; RH, right hindlimb.

Specimen	Slope	Standard Error of the Slope	r Value
Horse 1 LF	−0.012	9.13 × 10^−5^	−0.16 *
Horse 1 RF	excluded	excluded	excluded
Horse 1 LH	−0.010	7.13 × 10^−5^	−0.23 **
Horse 1 RH	−0.011	7.20 × 10^−5^	−0.24 **
Horse 2 LF	−0.0098	6.87 × 10^−5^	−0.18 *
Horse 2 RF	−0.0094	6.16 × 10^−5^	−0.29 **
Horse 2 LH	−0.0094	5.83 × 10^−5^	−0.29 **
Horse 2 RH	−0.0094	5.53 × 10^−5^	−0.35 **
Horse 3 LF	−0.0090	6.87 × 10^−5^	−0.30 **
Horse 3 RF	−0.0090	6.27 × 10^−5^	−0.34 **
Horse 3 LH	−0.0094	5.41 × 10^−5^	−0.33 **
Horse 3 RH	−0.0087	5.49 × 10^−5^	−0.43 ***
Horse 4 LF	−0.0086	5.89 × 10^−5^	−0.36 **
Horse 4 RF	−0.0082	7.12 × 10^−5^	−0.29 **
Horse 4 LH	−0.0084	5.70 × 10^−5^	−0.32 **
Horse 4 RH	−0.0077	6.39 × 10^−5^	−0.30 **
Horse 5 LF	−0.0074	8.36 × 10^−5^	−0.22 **
Horse 5 RF	−0.0072	8.65 × 10^−5^	−0.24 **
Horse 5 LH	−0.0074	7.14 × 10^−5^	−0.28 **
Horse 5 RH	−0.0076	6.66 × 10^−5^	−0.30 **
Horse 6 LF	−0.0069	7.25 × 10^−5^	−0.31 **
Horse 6 RF	−0.0057	8.80 × 10^−5^	−0.14 *
Horse 6 LH	−0.0048	7.27 × 10^−5^	−0.32 **
Horse 6 RH	−0.0048	7.39 × 10^−5^	−0.40 ***

## Data Availability

The data presented in this study are available on request from the corresponding authors. The data are not yet publicly available due to ongoing analysis by our group.

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
