# Peer review of "Relationship between CT-Derived Bone Mineral Density and UTE-MR-Derived Porosity Index in Equine Third Metacarpal and Metatarsal Bones"

_animals, 2023, doi:10.3390/ani13172780_

Round 1
Reviewer 1 Report
Thank you for the hard work and time that you have put into this research. I think your work is very relevant to the industry and I look forward to reading what you discover if you move forward from this "proof of concept" paper into more specific location examination (ie. distal palmar/plantar condyles).
I have a couple of minor comments/questions requiring some clarification.
* Minor Typographical error Line 118: “in the, in the”
* Line 213-215: I find this opening sentence awkward and a little confusing after re-reading it a number of times – could you please rephrase it without the “hereby”? I think perhaps these need to be 2 separate sentences?
* You state that UTE-MR is used for assessment of fluid in cortical bone (Line 97) and yet you have used an ROI that includes the entire distal MC3/MT3. Why did you not separate out cortical vs trabecular bone for these measurements?
* The advantages of 3T MRI over the standing low field units are many but the latter are the more ubiquitous at this time. Is UTE-MRI something that has potential to be used in the low field systems as well and if so, could you please address this in your discussion so that readers understand it is not necessarily limited to 3T magnets.
* How confident are you that you are not disrupting the normal behaviour of the BW and PW water protons by freezing and then defrosting the tissues? Given that you are suggesting this UTE-MR has the potential for use in clinical patients, do you not need to first make sure the water is behaving the same way in freeze/thawed tissue as it would fresh specimens?
Your Simple Summary and Abstract are excellent. I found your introduction to be comprehensive and very easy to read however your discussion did not seem to flow as well. I had to re-read it a number of times to follow your argument, particularly the first 3 paragraphs. Consider editing the beginning of your discussion to make it a little ore robust.
Author Response
Thank you very much for taking the time to review our manuscript.
Thank you for the hard work and time that you have put into this research. I think your work is very relevant to the industry and I look forward to reading what you discover if you move forward from this "proof of concept" paper into more specific location examination (ie. distal palmar/plantar condyles).
- Thank you for the constructive review of our manuscript. We hope we have addressed your comments below.
I have a couple of minor comments/questions requiring some clarification.
* Minor Typographical error Line 118: “in the, in the”
- Corrected
* Line 213-215: I find this opening sentence awkward and a little confusing after re-reading it a number of times – could you please rephrase it without the “hereby”? I think perhaps these need to be 2 separate sentences?
- Re-phrased accordingly.
* You state that UTE-MR is used for assessment of fluid in cortical bone (Line 97) and yet you have used an ROI that includes the entire distal MC3/MT3. Why did you not separate out cortical vs trabecular bone for these measurements?
- A paragraph to explain this decision has been added to the discussion (lines 285-308)
* The advantages of 3T MRI over the standing low field units are many but the latter are the more ubiquitous at this time. Is UTE-MRI something that has potential to be used in the low field systems as well and if so, could you please address this in your discussion so that readers understand it is not necessarily limited to 3T magnets.
- Whilst not currently commercially available, UTE has been implemented in experimental low field systems (see reference 48 and 49) and may become available in the future.
- Information was added into the first paragraph of the discussion (lines 256-259)
* How confident are you that you are not disrupting the normal behaviour of the BW and PW water protons by freezing and then defrosting the tissues? Given that you are suggesting this UTE-MR has the potential for use in clinical patients, do you not need to first make sure the water is behaving the same way in freeze/thawed tissue as it would fresh specimens?
- Mixed report in the literature if freezing/ thawing had an effect on MRI
- No specific studies on UTE sequences, use of frozen and thawed cadaver specimens widely used
- Information added to discussion (lines 272-283)
Comments on the Quality of English Language
Your Simple Summary and Abstract are excellent. I found your introduction to be comprehensive and very easy to read however your discussion did not seem to flow as well. I had to re-read it a number of times to follow your argument, particularly the first 3 paragraphs. Consider editing the beginning of your discussion to make it a little ore robust.
- Thank you, discussion has been extensively edited.
Reviewer 2 Report
Dear authors,
the premise of the paper is very good, but discussions and conclusions are more of an additional introduction. in fact, topics are discussed in the discussions that do not support the results obtained. More like a review of what is in the literature than discussions.It was not explained why the two components of bone, cortical and trabecular, were not considered separately.I believe that it would be appropriate to implement the discussions by correlating the arguments with the results obtained.
Line 63: the statement “physiological subchondral bone pathology” is improper; change “pathology” with “changes”. A pathology cannot be physiological
Line 73: I partially disagree. Bone scan and MRI can provide informations about bone repetitive stress injury, even in early stage (i.e. Ramzan and Powell EVJ 2010;42)
Line 83: I think mentioning and making a reference to “fragility fracture” is mandatory; in fact, the system that you described, is widely used in human medicine in patients with osteoporosis. This might be useful to better explain the differences in the etiopathogenesis of fatigue fractures in horses and humans, as well as the differences between the results obtained in your study and those described in humans
Line 94: specify that they are human studies
Line 100: add references
Line 249: you said that human studies consider PI limited to the cortical, but you consider both spongiosa and cortical together. Why? What studies are there to support your choice? There is no clear mention of the results of your study.
Author Response
Thank you very much for taking the time to review our manuscript.
Please find the addressed comments attached.
Dear authors,
the premise of the paper is very good, but discussions and conclusions are more of an additional introduction. in fact, topics are discussed in the discussions that do not support the results obtained. More like a review of what is in the literature than discussions.It was not explained why the two components of bone, cortical and trabecular, were not considered separately.I believe that it would be appropriate to implement the discussions by correlating the arguments with the results obtained.
- Thank you for the constructive review of our manuscript. Reasons for considering the whole condyle have been added to the discussion, which has also been re-written.
Line 63: the statement “physiological subchondral bone pathology” is improper; change “pathology” with “changes”. A pathology cannot be physiological
- Changed as suggested.
Line 73: I partially disagree. Bone scan and MRI can provide informations about bone repetitive stress injury, even in early stage (i.e. Ramzan and Powell EVJ 2010;42)
- Scintigraphy and MRI added as in vivo modalities for indirect assessment of microdamage and early stress injury (lines 83-88)
Line 83: I think mentioning and making a reference to “fragility fracture” is mandatory; in fact, the system that you described, is widely used in human medicine in patients with osteoporosis. This might be useful to better explain the differences in the etiopathogenesis of fatigue fractures in horses and humans, as well as the differences between the results obtained in your study and those described in humans
- Reference to fragility fractures was made in the introduction and different age groups and etiopathogenesis has been included in the discussion (lines 90 and 316-322)
Line 94: specify that they are human studies
- Added ‘in humans” to emphasise the fact that these are human studies
Line 100: add references
- References added about the use of UTE MRI in bone imaging (references 34-36).
Line 249: you said that human studies consider PI limited to the cortical, but you consider both spongiosa and cortical together. Why? What studies are there to support your choice? There is no clear mention of the results of your study.
- Explanation added in discussion (lines 285-308)
Reviewer 3 Report
This is an interesting proof of principle study on a small number of limbs from Thoroughbred racehorses of variable age whose exercise history and lameness status was unknown.
The authors used CT and UTE MRI measurements to assess bone mineral density and a porosity index. It is a shame that there was no correlative post mortem results to document the architecture of the bones. There was also no biomechanical testing of the bones.
I freely admit that I know very little about UTE MRI and do not have time to read more about it, so I have to accept that the authors are using it, and interpreting the results, correctly.
There appeared to be quite large variability in the results and the correlation coefficients were small; it was not clear which were statistically significant and which were not. No attempt was made to qualitatively describe these correlation coefficients, for example as weak. This is potentially misleading to the reader.
The authors fail to discuss the limitations of their study. There is some repetition in the Introduction and the Discussion which should be removed.
Insufficient detail is included in the Materials and Methods to allow the study to be reproduced. More information about the imaging parameters and the planes in which MR images were acquired is needed.
What p value was selected for statistical significance?
The Discussion is quite long and needs a bit more focus on what the results mean to horses and what further studies are needed.
Line 37 please qualify the level of correlation - weak etc.
Line 38 please clarify if this was of statistical significance
Line 41 Keywords are designed to provide additional words for a search engine that are not in the title of the manuscript.
Lines 45 & 46 Increased in comparison with what?
Line 80 replace , with ; or split this sentence into 2 – it makes no sense as written
Line 97 & 98 references needed
Line 118 'in the' is repeated
Line 131 ' various time periods' is very vague
Line 133 was CT scanning performed on frozen or unfrozen limbs?
Line 134 fan beam or cone beam?
Line 142 abbreviations not defined
More details of scanning technique required - which planes? How were limbs positioned?
What sort of coli was used?
Were the limbs wrapped in any way?
Line 179 data were
Line 180 r values
At what level was significance set?
Please qualify your r-values - weak, moderate etc., providing a reference
Line 186 image processing
Line 190 Low r values – and quite large variability
Was this statistically significant - please give p value
Please discuss the variability in r values among limbs
Table 1 please indicate which values were statistically significant
Lines 213 -5 This sentence makes little sense as written. Please rewrite
In the first 2 paragraphs of the Discussion it needs to be clearer that the references are all to UK based studies; training and racing conditions are very different in other major racing countries such as the USA & Hong Kong.
Paragraph 3 is repetitive of the Introduction
Line 251 larger than what?
Line 286 ‘UTE MRI derived bone porosity measurements are an excellent indirect measure of bone microstructure’
In what species has this been verified by post mortem bone histomorphometry? Is this translatable to horses?
What are the limitations of your study? I think that there are many which need to be itemised.
Some references are capitalised & other are not – please check for consistency
Please provide authors' names - not ‘et al.’ – this is lazy – and does not give credit to the co-authors
Ref 5 Which chapter, which pages, which edition?
Who authored the chapter?
see report to authors
Author Response
Thank you very much for taking the time to review our manuscript.
Please find the addressed comments attached.
This is an interesting proof of principle study on a small number of limbs from Thoroughbred racehorses of variable age whose exercise history and lameness status was unknown.
The authors used CT and UTE MRI measurements to assess bone mineral density and a porosity index. It is a shame that there was no correlative post mortem results to document the architecture of the bones. There was also no biomechanical testing of the bones.
I freely admit that I know very little about UTE MRI and do not have time to read more about it, so I have to accept that the authors are using it, and interpreting the results, correctly.
- Thank you for the constructive review of our manuscript, we hope we have addressed your concerns below.
There appeared to be quite large variability in the results and the correlation coefficients were small; it was not clear which were statistically significant and which were not. No attempt was made to qualitatively describe these correlation coefficients, for example as weak. This is potentially misleading to the reader.
- Discussion of weak correlation and variability made added (lines 310-336)
The authors fail to discuss the limitations of their study. There is some repetition in the Introduction and the Discussion which should be removed.
- Discussion of limitations was added and the discussion has been re-written. (lines 338-354)
Insufficient detail is included in the Materials and Methods to allow the study to be reproduced. More information about the imaging parameters and the planes in which MR images were acquired is needed.
- Information added in the MR imaging section (lines 150-159)
What p value was selected for statistical significance?
- P=0.005, added to the statistics section. (line198/ 199)
The Discussion is quite long and needs a bit more focus on what the results mean to horses and what further studies are needed.
- We agree, the discussion has been re-written.
Line 37 please qualify the level of correlation - weak etc.
- Correlation specified as “weak”
Line 38 please clarify if this was of statistical significance
- Rephrased to make it clearer that statistical significance was reached.
Line 41 Keywords are designed to provide additional words for a search engine that are not in the title of the manuscript.
- Key words have been adjusted as suggested.
Lines 45 & 46 Increased in comparison with what?
- Information added as requested (“general horse population”)
Line 80 replace , with ; or split this sentence into 2 – it makes no sense as written
- Semicolon added
Line 97 & 98 references needed
- References added about the use of UTE MRI in bone imaging (references number 34-36)
Line 118 'in the' is repeated
- Corrected
Line 131 ' various time periods' is very vague
- Information added as requested (storage period ranged from 40 days to 282 days, line 139).
Line 133 was CT scanning performed on frozen or unfrozen limbs?
- CT scanning was performed on unfrozen limbs. This information was added as suggested (line 140/ 141).
Line 134 fan beam or cone beam?
- Information adjusted (“A 64-slice fan beam CT was used).
Line 142 abbreviations not defined
More details of scanning technique required - which planes? How were limbs positioned?
What sort of coli was used?
Were the limbs wrapped in any way?
- Definition of the abbreviations TE and TR added
- More detail provided as requested (line 150-159)
- Isotropic voxels
- Fat suppression
- Matrix size and FOV
- Frontal plane
- Knee coil (15 channels, Tx/ Rx)
- Wrapped in plastic specimen bag
- Positioning of limbs: dorsal side up, hoof pointing outwards (“head first supine”)
Line 179 data were
- Corrected
Line 180 r values
At what level was significance set?
Please qualify your r-values - weak, moderate etc., providing a reference
- Information added: “A p-value of less than 0.05 was considered statistically significant.”
- R values classified and reference provided (also included in Table 1)
Line 186 image processing
- Corrected
Line 190 Low r values – and quite large variability
Was this statistically significant - please give p value
- Paragraph about statistical significance added from (line 207 to 210)
Please discuss the variability in r values among limbs
- Added to discussion (line 324-336)
Table 1 please indicate which values were statistically significant
- Requested information was added to the table legend (line 233- 238)
- “For all specimens the correlation coefficient (r value) was statistically significant different from 0 with a p value of less than 0.00001.”
Lines 213 -5 This sentence makes little sense as written. Please rewrite
- Re-written as suggested
In the first 2 paragraphs of the Discussion it needs to be clearer that the references are all to UK based studies; training and racing conditions are very different in other major racing countries such as the USA & Hong Kong.
- Clarification that these are UK based studies added
Paragraph 3 is repetitive of the Introduction
- Discussion was re-written taking this comment into account.
Line 251 larger than what?
- Specified: “much larger amount of trabecular bone than the diaphysis”
Line 286 ‘UTE MRI derived bone porosity measurements are an excellent indirect measure of bone microstructure’
In what species has this been verified by post mortem bone histomorphometry? Is this translatable to horses?
- Clarification that this has been verified in a human study has been added (line 266/ 267)
- Reference to similar osseous microstructure in mammalian species has been added (reference number 52 and 53, lines 265-271)
What are the limitations of your study? I think that there are many which need to be itemised.
- A paragraph on the limitations of the study has been added to the discussion (lines 338-354)
- Small sample size
- No training records
- Post mortem
- No biomechanical testing
- Further ROI segmentation
Some references are capitalised & other are not – please check for consistency
- Revised as requested.
Please provide authors' names - not ‘et al.’ – this is lazy – and does not give credit to the co-authors
- Citation style adjusted accordingly.
Ref 5 Which chapter, which pages, which edition?
Who authored the chapter?
- Information added as requested.
Round 2
Reviewer 2 Report
Dear authors,
I really appreciate the changes that were made as a result of my previous comments. In my opinion, discussion and conclusions are now well supported by the results. Thank you for your work.
Kind regards
Author Response
Thank you very much for taking the time to review our manuscript. Your input is much appreciated.
Reviewer 3 Report
The revised version of this manuscript is vastly superior to the originally submitted version. The methodology is now much more complete. The Discussion is much more relevant and useful. My only comment is for Line 209- I clicked on the link for 47, out of passing interest, and it did not appear to define the strength of r values!
Author Response
Thank you very much for your review. Your input is much appreciated.
We have updated the link according to your suggestions. The relevant paragraph on the website is the last paragraph of the section "Calculator procedure".